# Subcortical correlates of consciousness with human single neuron recordings

Michael Pereira[1,2,3]*[†], Nathan Faivre[1,4][†], Fosco Bernasconi[1][†], Nicholas Brandmeir[5], Jacob E Suffridge[2,6], Kaylee Tran[2], Shuo Wang[2,6,7], Victor Finomore[2], Peter Konrad[5], Ali Rezai[2,5][‡], Olaf Blanke[1,8][‡]

[1]Laboratory of Cognitive Neuroscience, Neuro-X Institute & Brain Mind Institute, Faculty of Life Sciences, Swiss Federal Institute of Technology (EPFL), Geneva, Switzerland; [2]Department of Neurosciences, WVU Rockefeller Neuroscience Institute, Morgantown, United States; [3]University Grenoble Alpes, Inserm U1216, CHU Grenoble Alpes, Grenoble Institut Neurosciences, Grenoble, France; [4]University Grenoble Alpes, University Savoie Mont Blanc, CNRS, LPNC, Grenoble, France; [5]Departments of Neurosurgery, WVU Rockefeller Neuroscience Institute, Morgantown, United States; [6]Department of Computer Science and Electrical Engineering, West Virginia University, Morgantown, United States; [7]Department of Radiology, Washington University in St. Louis, St. Louis, United States; [8]Department of Clinical Neurosciences, University Hospital Geneva, Geneva, Switzerland

*For correspondence:
michael.pereira@univ-grenoble-alpes.fr

[†]These authors contributed equally to this work
[‡]These authors also contributed equally to this work

## eLife Assessment

This **important** study reports human single-neuron recordings in subcortical structures while participants performed a tactile detection task around the perceptual threshold. The study and the analyses are well conducted and provide **convincing** evidence that the thalamus and the subthalamic nucleus contain neurons whose activity correlates with the task, with stimulus presentation, and even with whether the stimulation is consciously detected or not. The study will be relevant for researchers interested in the role of subcortical structures in tactile perception and the neural correlates of consciousness.

**Abstract** Subcortical brain structures such as the subthalamic nucleus or the thalamus are involved in regulating motor and cognitive behavior. However, their contribution to perceptual consciousness remains unclear, due to the inherent difficulties of recording subcortical neuronal activity in humans. Here, we asked neurological patients undergoing surgery for deep brain stimulation to detect weak vibrotactile stimuli applied on their hand while recording single neuron activity from the tip of a microelectrode. We isolated putative single neurons in the subthalamic nucleus and thalamus. A significant proportion of neurons modulated their activity while participants were expecting a stimulus. We found that the firing rate of 23% of these neurons differed between detected and undetected stimuli. Our results provide direct neurophysiological evidence of the involvement of the subthalamic nucleus and the thalamus for the detection of vibrotactile stimuli, thereby calling for a less cortico-centric view of the neural correlates of consciousness.

## Introduction

Current methods to investigate the *neural correlates of consciousness* aim at contrasting the neural activity associated with different percepts under constant sensory stimulation to identify the minimal

set of neuronal events sufficient for a specific conscious percept to occur (*Koch et al., 2016*; *Seth and Bayne, 2022*). Typically, this involves asking participants to report whether a stimulus with an intensity around detection threshold is present or not. Taking advantage of the wealth of invasive electrophysiology recordings available, researchers have documented such correlates with detection tasks in rodents (*Schmack et al., 2021*), birds (*Nieder et al., 2020*), and nonhuman primates (*de Lafuente and Romo, 2006*). However, the use of animal models to study consciousness raises specific ethical concerns (*Mazor et al., 2023*) and requires interpreting behavioral responses with caution (*Birch et al., 2022*). Research into the neural correlates of consciousness in human volunteers is enriched by the analysis of fine-grained subjective reports to rule out various confounds (e.g. attention, memory, report), but suffers from less spatially and temporally resolved physiological measurements. Indeed, only very few studies have found such correlates at the single neuron level (*Fried et al., 1997*; *Gelbard-Sagiv et al., 2018*; *Pereira et al., 2021*; *Quiroga et al., 2008*; *Reber et al., 2017*) and only in cortical regions. The role of subcortical structures for perceptual consciousness is theoretically relevant (*Aru et al., 2020*; *Dehaene and Changeux, 2011*; *Schiff, 2008*; *Seth and Bayne, 2022*; *Ward, 2013*) with some empirical support from detection studies in nonhuman primates (*Haegens et al., 2014*; *Tauste Campo et al., 2019*; *Vázquez et al., 2012*; *Vázquez et al., 2013*), as well as functional imaging or local field potentials in humans (*Kronemer et al., 2022*; *Levinson et al., 2021*). Nonetheless, it remains unknown how the firing rate of subcortical neurons changes when a stimulus is consciously perceived. Single-unit recordings provide a much higher temporal resolution than functional imaging, which helps assess how the neural correlates of consciousness unfold over time. Contrary to local field potentials, single-unit recordings can expose the variety of functional roles of neurons within subcortical regions, thereby offering a potential for a better mechanistic understanding of perceptual consciousness. Here, we recorded individual neurons from the subthalamic nucleus (STN) and thalamus of human participants during 36 deep brain stimulation surgeries. Participants detected vibrotactile stimuli provided at the perceptual threshold, and we tested how neurons in both subcortical structures were modulated by the task, the onset of the stimulus, or the detection or not of the stimulus.

## Results

### Task and behavior

Deep brain stimulation surgeries provide a unique opportunity to record the activity of single neurons in subcortical structures of the human brain. Microelectrode recordings are performed routinely after patients are awakened from anesthesia, to allow electrophysiologists and neurosurgeons to identify the target brain structure along the planned trajectory (*Figure 1B*, *Figure 1—figure supplement 1*). During this procedure, we attached a vibrotactile stimulator to the palm of the hand contralateral to the microelectrode recordings and estimated the stimulus intensity corresponding to participants' individual tactile detection threshold. Once stable neuronal activity could be recorded in the target brain region (STN or thalamus), we proceeded to the main experiment, which comprised one or two sessions of 71 trials (total: 48 sessions). Each trial started with an audio 'go' cue, followed by a vibrotactile stimulus applied at any time between 0.5 and 2.5 s after the end of the cue (i.e. stimulation window), except for 20% of catch trials in which no stimulus was applied (*Figure 1A*). After a random delay ranging from 0.5 to 1 s, a 'respond' cue was played, prompting participants to verbally report whether they felt a vibration or not. Therefore, none of the reported analyses are confounded by motor responses. Using a staircase procedure, the stimulus intensity was kept around the detection threshold over the whole experiment. When possible, participants were trained to perform the task prior to the surgery.

When analyzing tactile perception, we ensured that our results were not contaminated by fluctuating attention and arousal due to the surgical procedure. Based on objective criteria, we excluded a specific series of trials from analyses and focused on trials where hits and misses occurred in commensurate proportions (see Methods). This procedure led us to keep 36 sessions out of 48 with a mean of 24.0 [95% confidence interval = 22.0, 25.9] hit trials and 22.7 [20.8, 24.5] miss trials. Permutation tests at the single-participant level indicated that detected and missed stimuli were of similar intensity, except in 5 sessions for which the intensity of detected stimuli was higher on average. Likewise, detected and missed stimuli had similar onsets, except in 1 session for which stimuli with late onsets

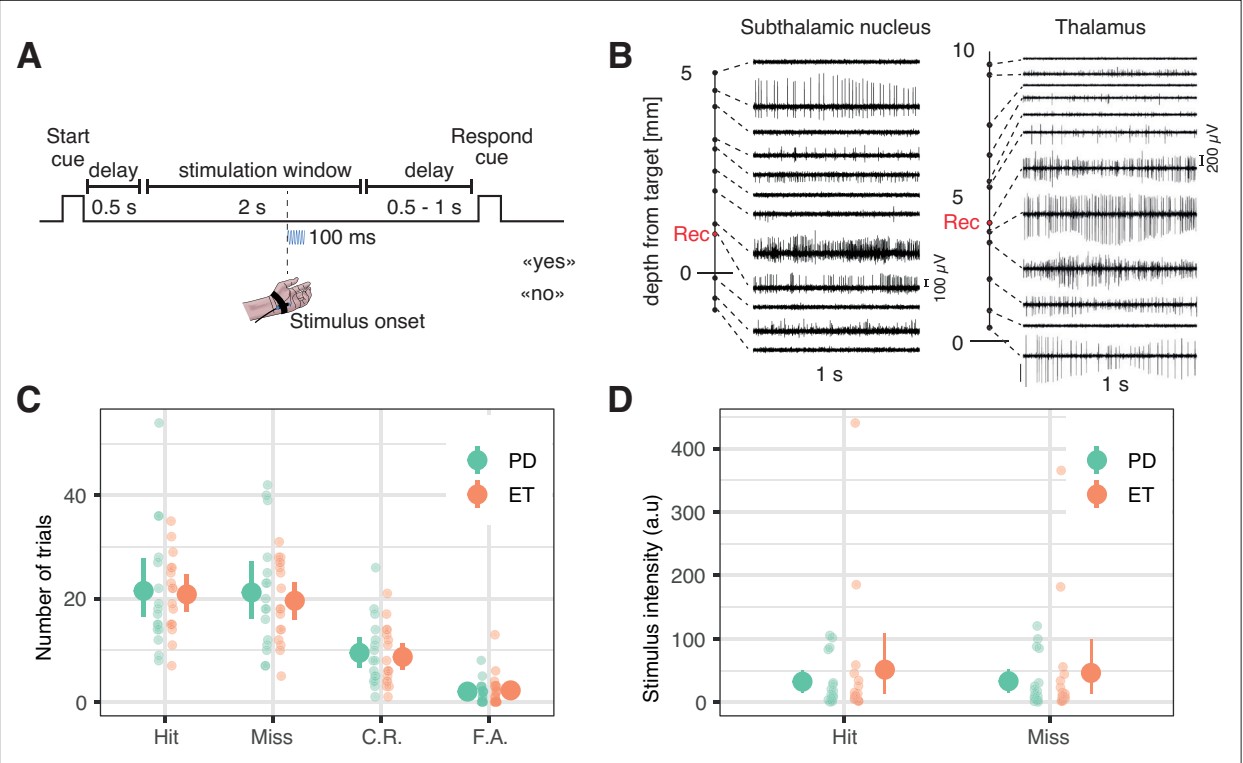

**Figure 1.** Task and behavior. (**A**) Task timeline. Each trial started with an auditory start cue, followed by a 0.5 s delay. Next, the stimulus could occur anytime during a 2 s stimulation window. After a variable 0.5–1 s delay, a response cue prompted patients to answer whether or not they detected the stimulus. (**B**) Two example sets of 1-s-long microelectrode recordings along the surgical tract showing specific firing for the subthalamic nucleus (left) and the motor thalamus (right). The depth at which the research data was collected is represented as a red dot (see *Figure 1—figure supplement 1* for anatomical correspondence). (**C**) Number of hits, misses, correct rejections (C.R.), and false alarms (F.A.) collected during the main experiment. (**D**) Averages of the absolute vibrotactile intensity in hits and misses in arbitrary units (values cannot be compared between participants). In panels C and D, each small dot represents a participant with Parkinson's disease (PD, in green, n = 16) or essential tremor (ET, in orange, n = 17). Big dots represent averages; error bars represent 95% *confidence intervals*.

The online version of this article includes the following figure supplement(s) for figure 1:

**Figure supplement 1.** Confirmatory localization analysis.

**Figure supplement 2.** Hit rate and false alarm rate observed during the main experiment and the training session.

were predominantly missed, and in 2 sessions for which stimuli with early onsets were predominantly missed. The hit rate was comparable between participants with Parkinson's disease (0.51 [0.49, 0.53]) and essential tremor (0.52 [0.51, 0.53], Wilcoxon rank sum test: W=114.5, p=0.45; *Figure 1C*). Catch trials were separated into 9.1 [8.1 10.1] correct rejections and 2.1 [1.7, 2.6] false alarms, with an equivalent false alarm rate between participants with Parkinson's disease (0.24 [0.19, 0.28]) and essential tremor (0.24 [0.18, 0.30], Wilcoxon rank sum test: W=145, p=0.76). Intraoperative behavior was similar to the behavior observed during the training session (*Figure 1—figure supplement 2*) and similar to what we found recently in a cohort of healthy participants using the same task (*Pereira et al., 2021*).

## Neuronal firing was modulated by the task

We performed a total of 48 (STN: 25, Thal: 23) successful microelectrode recording sessions during 36 surgeries for deep brain stimulation electrode implantation. We isolated 50 putative single neurons (STN: 26, Thal: 24) according to spike sorting metrics (*Figure 2—figure supplement 1A–G*). We ensured that all neurons showed stable spike amplitudes during the recording (*Figure 2—figure supplement 1H–J*). We also ensured that for every analysis, a minimum of 20 trials per condition were kept after removing artifacts. First, we looked for cue-selective neurons that modulate their firing rate during the 500 ms delay following the end of the 'go' cue, compared to a 500 ms pre-cue baseline period. There were 8/44 (18 %) cue-selective neurons (*Figure 2A*; 6 neurons were removed from the

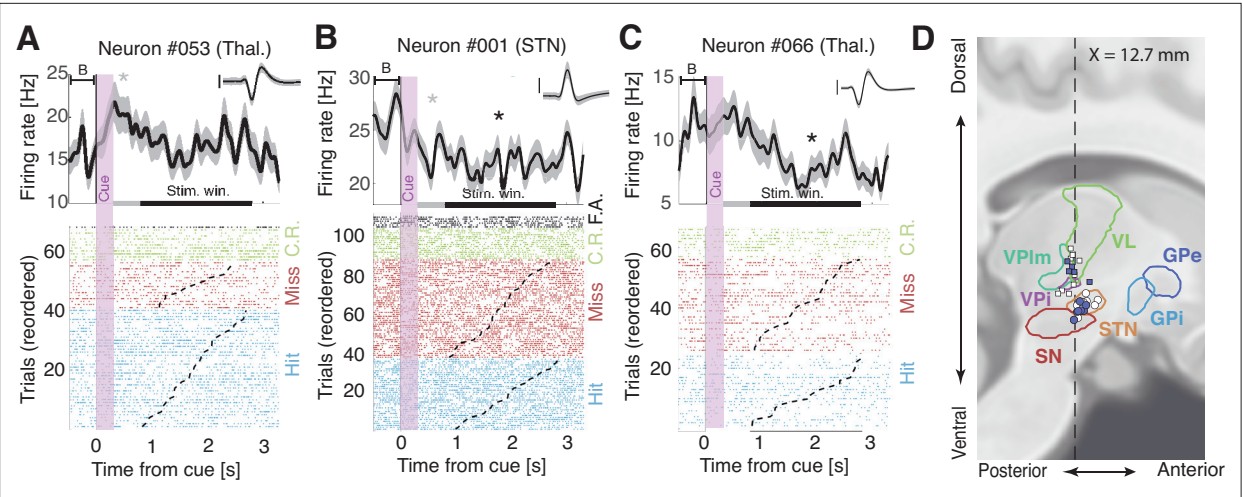

**Figure 2.** Representative cue- and task-responsive neurons in distinct patients. (**A–C**) Upper panels: firing rates time-locked to the onset of a trial (300-ms-long auditory cue; vertical purple shade), compared to a 500 ms pre-cue baseline (B). Two significance windows were tested: the post-cue window (500 ms after cue offset; gray horizontal bar; cue-selective neurons) or the stimulation window (800–2800 ms post-cue; black horizontal bar; task-selective neurons). Asterisks represent statistical significance (p<0.05). Shaded areas indicate bootstrapped standard errors. Inset: corresponding action potentials (shaded area indicates standard deviation; vertical bar corresponds to 100 μV, duration: 2.5 ms). Lower panels: raster plot with trials sorted by stimulus onset (dashed lines) and type: hits (blue), misses (red), correct rejections (C.R.; green), and false alarms (F.A.; black). (**A**) Cue-selective neuron in the thalamus. (**B**) Cue- and task-selective neurons in the STN. (**C**) Task-selective neuron in the thalamus. (**D**) Sagittal view of recording locations for thalamic (squares) and subthalamic (circles) targets (see *Figure 2—figure supplement 2* for a coronal view) for patients for which we could obtain anatomical images. Filled circles or squares are cue/task-selective neurons. Legend: VL: ventral lateral thalamus, VPlm: ventral posterior lateral and medial thalamus, VPi: ventral posterior inferior thalamus, STN: subthalamic nucleus, SN: substantia nigra, GPi/e: globus pallidus internalis/externalis.

The online version of this article includes the following figure supplement(s) for figure 2:

**Figure supplement 1.** Spike sorting metrics.

**Figure supplement 2.** Coronal view of recording locations.

analysis due to an insufficient number of trials). We confirmed that these 8 cue-selective neurons could not have been obtained by chance by comparing this number to a null distribution obtained by permuting the sign of the trial-wise differences 1000 times (sign permutation test: p<0.001). The proportion of cue-selective neurons was not significantly different in the STN (21%) and thalamus (15%; difference: p=0.31, sign permutation test), and 6 out of 8 neurons showed a decrease in firing rate compared to the pre-cue baseline.

Next, we investigated how many neurons showed task-selective modulations by comparing firing rates during the 2 s stimulation window to the 500 ms pre-cue baseline, indicating a modulation of their firing rate when a stimulus is expected. There were 9/44 (20%) task-selective neurons (sign permutation test: p=0.003) with a similar proportion in the STN (21%) and thalamus (20%; *Figure 2B–D*). Interestingly, 8 out of 9 neurons decreased their firing rate relative to the pre-cue baseline. In both regions, a significant proportion (44%; permutation test: p<0.001) of the task-selective neurons were also cue-selective, modulating their firing rate before any sensory stimulation necessary for a decision occurred. Therefore, these cue- and task-selective neurons are unlikely to be involved in decision-related action selection or cancellation (*Bastin et al., 2014*; *Mosher et al., 2021*) but should be involved in the detection task per se.

## Neuronal firing was modulated by the stimulus

We then searched for neurons that modulate their firing rate after the stimulus onset compared to a 300 ms pre-stimulus baseline while correcting for possible drifts in the firing rate during the trial (see Methods). We found 8/37 such stimulus-selective neurons (22%, sign permutation test: p=0.011; *Figure 3A–D*; 13 neurons were removed due to an insufficient number of trials), with 29% in the STN and 15% in the thalamus. These differences occurred 211 ms±34 after the stimulus onset, lasted for an average of 126 ms±28, and 6 out of 8 neurons showed a decrease in firing rate after the stimulus onset. These results show that subthalamic and thalamic neurons are modulated by stimulus

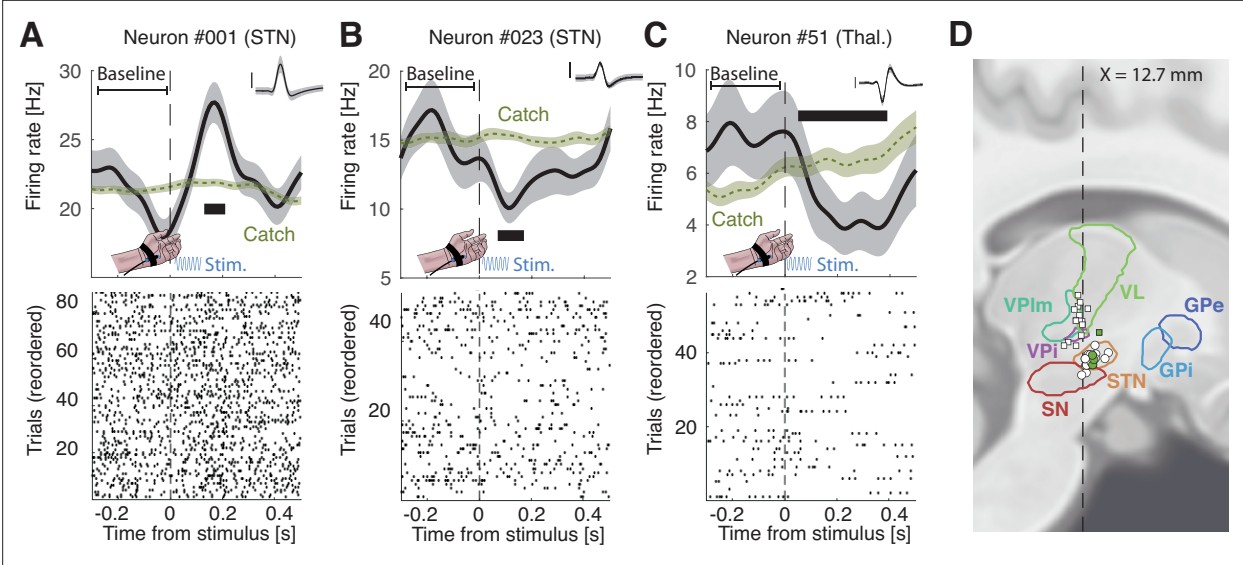

**Figure 3.** Representative stimulus-responsive neurons in distinct patients. (**A–C**) Upper panels: firing rate time-locked to the onset of the stimulus (100 ms vibrotactile stimulation; blue sinusoid) for all trials before baseline correction. Green trace represents corresponding activity for catch trials. Thick horizontal black segments show significant time windows. Shaded areas indicate bootstrapped standard errors. Inset: corresponding action potentials (shaded area indicates standard deviation; vertical bar corresponds to 100 μV, duration: 2.5 ms). Lower panels: raster plot. The 300 ms pre-stimulus baseline was used only for statistics. (**D**) Sagittal view of recording locations for thalamic (squares) and subthalamic (circles) targets (see *Figure 3—figure supplement 1* for a coronal view) for patients for which we could obtain anatomical images. Filled circles or squares are sensory-selective neurons. Legend: VL: ventral lateral thalamus, VPlm: ventral posterior lateral and medial thalamus, VPi: ventral posterior inferior thalamus, STN: subthalamic nucleus, SN: substantia nigra, GPi/e: globus pallidus internalis/externalis.

The online version of this article includes the following figure supplement(s) for figure 3:

**Figure supplement 1.** Coronal view of recording locations.

onset, irrespective of whether it was reported or not, even though no immediate motor response was required.

## Neuronal firing was modulated by tactile perception

Having identified subcortical neurons that were cue-, task-, or stimulus-selective, we next sought to assess the role of these structures in conscious detection by comparing firing rates time-locked to detected vs missed stimuli. Of the 50 neurons recorded, 35 were associated with periods of high-quality behavior, allowing us to assume tactile stimulation at the perceptual threshold. We found 8 neurons (23%) showing a significant difference between hits and misses after stimulus onset (trial permutation test: p=0.0060; *Figure 4A–D*, *Figure 4—figure supplement 1*). Each neuron was found in a different participant. The proportion of these perception-selective neurons was similar in the STN (27%) and the thalamus (20%; difference: p=0.529; trial permutation test). These differences in firing rates occurred on average 160 ms±36 after the stimulus onset and lasted for an average of 100 ms±10. We note that the timings of sensory and perception effects in *Figures 3 and 4* showed a bimodal distribution with an early cluster (149 ms for sensory neurons; 121 ms for perception neurons; c.f. Methods) and a late cluster (330 ms for sensory neurons; 315 ms for perception neurons; *Figure 5*). We note that 6 out of 8 neurons had higher firing rates for missed trials than hit trials. None of the aforementioned neurons showed sustained differences between the highest and lowest stimulus amplitudes nor between early and late stimulus onset within the 2 s stimulus window (*Figure 4—figure supplement 3*). Our control analyses confirm that our results do not stem from slight differences in stimulus amplitudes due to the staircase procedure or spurious differences induced by the start or response cues. Qualitatively, we found little overlap between task-, stimulus-, and perception-selective neurons (one neuron selective for all three categories, one task- and perception-selective neuron, two stimulus- and perception-selective neurons, and one task- and perception-selective neuron; *Figure 4—figure supplement 4*). This result suggests that these two subcortical structures display neurons with several different

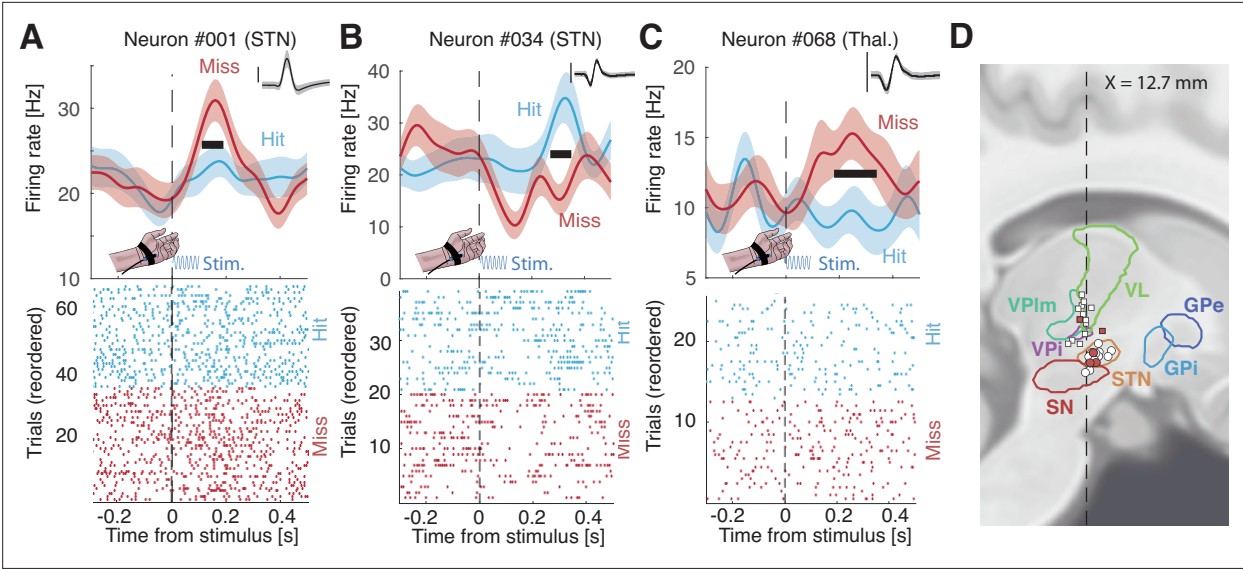

**Figure 4.** Representative perception-selective neurons in distinct patients. (**A–C**) Upper panels: firing rate time-locked to the onset of the stimulus (100 ms vibrotactile stimulation; blue sinusoid) for hits (light blue) and misses (red). Thick horizontal black segments show significant time windows. Shaded areas indicate bootstrapped standard errors. Inset: corresponding action potentials (shaded area indicates standard deviation; vertical bar corresponds to 100 μV, duration: 2.5 ms). Lower panels: raster plot for hits (light blue) and misses (red). See *Figure 4—figure supplement 1* for all neurons that are perception- or sensory-selective. (**D**) Sagittal view of recording locations for thalamic (squares) and subthalamic (circles) targets (see *Figure 4—figure supplement 2* for a coronal view) for patients for which we could obtain anatomical images. Filled circles or squares are perception-selective neurons. Legend: VL: ventral lateral thalamus, VPlm: ventral posterior lateral and medial thalamus, VPi: ventral posterior inferior thalamus, STN: subthalamic nucleus, SN: substantia nigra, GPi/e: globus pallidus internalis/externalis.

The online version of this article includes the following figure supplement(s) for figure 4:

**Figure supplement 1.** Firing rate time-locked to the onset of the stimulus (100 ms vibrotactile stimulation; vertical dashed line) for hits (light blue), misses (red), and for combined data (both; black).

**Figure supplement 2.** Coronal view of recording locations.

**Figure supplement 3.** Neurons from *Figure 4* for different stimulus intensities and onsets.

**Figure supplement 4.** Distribution of subthalamic neurons and relation to beta oscillations.

functional roles. We also found no clear indication that neurons with a beta-band oscillatory component were more or less selective.

## Discussion

The importance of cortico-subcortical loops for physiological and cognitive functions is well established (*Shepherd and Yamawaki, 2021*). Yet, while the role of subcortical structures in perceptual consciousness is largely acknowledged (*Aru et al., 2020*; *Dehaene and Changeux, 2011*; *Koch et al., 2016*; *Shepherd and Yamawaki, 2021*; *Ward, 2013*), it remains poorly described in humans. This limit is likely due to the difficulty of recording subcortical activity in awake humans capable of providing conscious reports under controlled experimental conditions. We report the first intraoperative recordings of subcortical neurons in awake individuals during a detection task. By imposing a delay between the end of the tactile stimulation window and the subjective report, we ensured that neuronal responses reflected stimulus detection and not mere motor responses. In addition, because stimuli were applied on the palm, we asked participants to provide detection responses orally to avoid confounding neural activity related to sensory and motor processes of the upper limb. Our main result is that the activity of subcortical neurons covaries with subjective reports following the presentation of detected vs missed tactile stimuli. This result confirms that the neuronal underpinnings of tactile detection can be observed at the scale of single neurons in humans (*Fried et al., 1997*; *Gelbard-Sagiv et al., 2018*; *Pereira et al., 2021*; *Quiroga et al., 2008*; *Reber et al., 2017*) but also shows for the first time that they are not limited to the cortex.

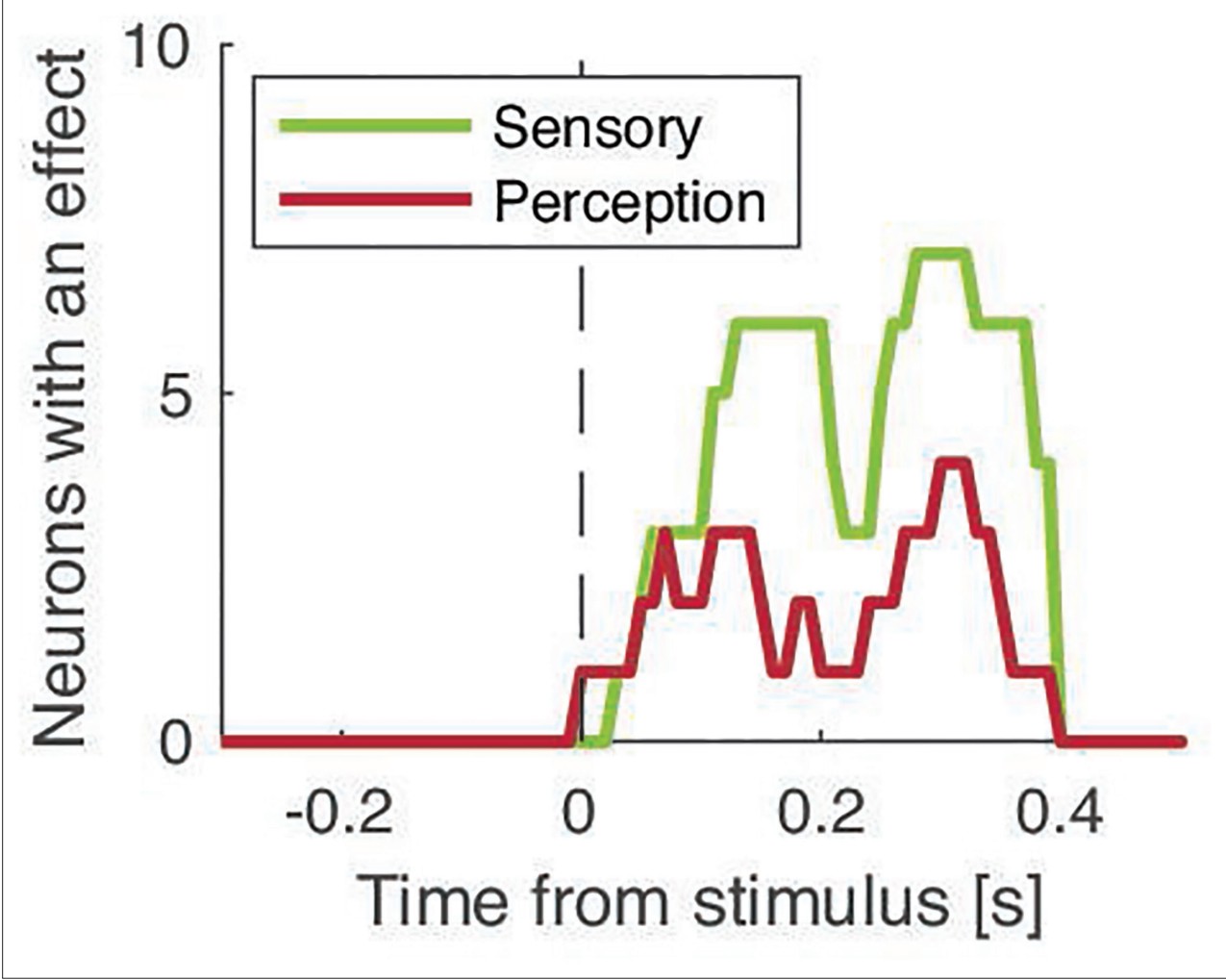

**Figure 5.** Number of neurons with a significant effect over time for sensory-selective neurons (green) and perception-selective neurons (red). Arrows show the corresponding early and late cluster centroids (green for sensory; red for perception).

Our findings that neurons in the thalamus modulate their activity according to tactile detection add to the existing evidence in favor of the role of the thalamus for perceptual consciousness. Indeed, thalamic activity and, more precisely, thalamocortical loops are often considered key to gate sensory stimuli to conscious access (*Ward, 2013*). In nonhuman primates, for example, oscillatory thalamic activity predicts tactile detection (*Haegens et al., 2014*), and functional interactions between the somatosensory thalamus and the cortex increase when a tactile stimulus is detected (*Tauste Campo et al., 2019*). In humans, thalamic local field potentials and fMRI activity were higher for seen vs unseen stimuli (*Kronemer et al., 2022*; *Levinson et al., 2021*), and causal effects of thalamic stimulation on the levels of consciousness have been found (*Schiff et al., 2007*). Future studies with higher neuronal yields will be helpful in assessing the contribution of distinct thalamic territories to tactile consciousness, focusing notably on the ventral caudal part, which contains neurons with tactile receptive fields. Moreover, given the prominent role of the thalamus in multisensory processing (*Yang et al., 2025*), it will be interesting to assess if it is specifically involved in tactile consciousness or if it has a supramodal contribution, akin to what is found in the cortex (*Filimonov et al., 2022*; *Noel et al., 2018*; *Sanchez et al., 2020*).

Concerning the STN, a possibility is that perception-selective neurons determine stimulus detection through the regulation of decisional processes. Indeed, previous studies reported a modulatory role of subthalamic activity on decisional processes, notably by elevating the decisional threshold on accumulated sensory evidence (*Bogacz and Gurney, 2007*; *Cavanagh et al., 2011*; *Green et al., 2013*; *Herz et al., 2016*). In a recent study in which we measured the activity of cortical neurons in a

similar task, we showed that evidence accumulation is also at play during conscious detection (*Pereira et al., 2021*). Based on this finding, we proposed that percepts fade in and out of consciousness when evidence accumulated by cortical neurons passes a given threshold (*Pereira et al., 2022*). The present results, therefore, indicate that the contribution of subthalamic neurons to decisional processes is not limited to discrimination tasks or motor planning, but may also regulate the threshold at which accumulated evidence gives rise to a conscious percept. Considering the inhibitory role of the STN on the cortex (*Mink, 1996*), the fact that many of the perception-selective neurons we found had higher firing rates for misses than for hits suggests a role in elevating that threshold, similar to what is found in decision tasks manipulating conflict or cautiousness and requiring immediate responses (*Bastin et al., 2014*; *Cavanagh et al., 2011*; *Frank et al., 2007*; *Herz et al., 2016*; *Mosher et al., 2021*). Thus, our results suggest that the STN plays an important role in a subcortical network gating conscious access, although it might not encode conscious content per se (*Aru et al., 2012*).

Apart from perception-selective neurons, we also found a distinct population of neurons in both the STN and thalamus that modulated their firing rate both after the cue and during the task, and therefore much before the stimulus onset. These neurons cannot be involved in detection-related processes but could instead be involved in task switching (*Hikosaka and Isoda, 2010*; *Laquitaine et al., 2024*). Microelectrode recordings could therefore reconcile the multiple roles ascribed to subcortical structures by exposing the variety of functional roles of the neurons inside those regions. We also found neurons that modulated their firing rates after the stimulus onset, irrespective of detection, similar to animal works in the STN (*Al Tannir et al., 2023*; *Coizet et al., 2009*) and thalamus (*Tauste Campo et al., 2019*; *Vázquez et al., 2012*). Our results should be taken with caution as they are based on a small number of neurons due to the high complexity of intraoperative recordings, and because the number of trials we could collect was not sufficient to test the computational mechanisms underlying the neuronal activity we recorded. Interestingly, we found that both sensory and perception effects could be grouped into an early and a late cluster. The early cluster's average timing around 150 ms post-stimulus corresponds to the onset of a putative cortical correlate of tactile consciousness, the somatosensory awareness negativity (*Dembski et al., 2021*). Similar electroencephalographic markers are found in the visual and auditory modality. It is unclear, however, whether these markers are related to perceptual consciousness or selective attention (*Dembski et al., 2021*). The late cluster is centered around 300 ms and could correspond to a well-known electroencephalographic marker, the P3b (*Polich, 2007*), whose association with perceptual consciousness has been questioned (*Dembski et al., 2021*; *Pitts et al., 2014*), although brain activity related to consciousness has been observed at similar timing even in the absence of report demands (*Sergent et al., 2021*; *Stockart et al., 2024*). It is also important to note that these clusters contain neurons with both increased and decreased firing rates following the stimulus onset, similar to what was observed previously in the posterior parietal cortex (*Pereira et al., 2021*). Future studies ruling out the presence of motor preparation triggered by perceived stimuli (*Bennur and Gold, 2011*; *Fang et al., 2024*; *Twomey et al., 2016*) and verifying that similar neuronal activity occurs in the absence of task demands (no-reports; *Tsuchiya et al., 2015*) or attention (*Wyart and Tallon-Baudry, 2008*) will be useful to support that subcortical neurons contribute specifically to perceptual consciousness.

In sum, our study provides neurophysiological evidence from single neurons in humans that subcortical structures play a significant role in tactile detection either by themselves (*Ward, 2013*) or through their numerous connections with the cortex (*Dehaene and Changeux, 2011*). A comprehensive account of the neural correlates of consciousness should, therefore, not be cortico-centric but also consider subcortical contributions.

## Methods

### Participants

We recorded high-impedance electrophysiological signals from microelectrodes inserted intraoperatively in the STN of 32 participants with Parkinson's disease or essential tremor undergoing deep brain stimulation electrode implantation surgeries (N=36; 4 participants had two surgeries, one for each side; convenience sampling). For individuals with Parkinson's disease, the age at the time of the recording was 60.4±2.7 years, and the average UPDRS III score was 40.6±3.0 prior to surgery and was reduced to 20.8±2.8 after the surgery (p=0.0015, z=3.18). We also recorded intraoperatively in the

thalamus of individuals with essential tremor undergoing deep brain stimulation surgeries. The age at the time of the recording was 68.9±3.2 years, and the average TETRA motor score was 20.1±2.9 prior to surgery. The study was approved by the institutional review board of the West Virginia University Hospital (WVU02HSC17; #1709745061), and all participants provided written informed consent prior to any data collection.

## Experimental procedure

Participants performed a tactile detection task programmed in MATLAB using the Psychophysics toolbox (*Brainard, 1997*; *Kleiner et al., 2007*; *Pelli, 1997*). When possible, participants were trained a few days before the surgery (N=18/36 surgeries). Participants sat in a reclining chair in a quiet room (training session) or were lying in the operating room (main session). Every trial started with a 300-ms-long auditory 'go' cue delivered through an external loudspeaker placed near the participants. Following the end of the go cue and a delay of 500 ms, a 100 ms vibrotactile stimulus could be delivered at any time during a 2 s stimulation window (i.e. uniform distribution between 0.8 and 2.8 s after the onset of the go cue; *Figure 1A*) on the lateral palm contralateral to the deep brain implant. Stimuli were applied using an MMC3 Haptuator vibrotactile device from TactileLabs Inc (Montréal, Canada) driven by a 230 Hz sinusoid audio signal. Participants reported orally whether they felt the stimulus or not and whether they were confident in their answer or not after an auditory 'respond' cue played 1 s after the end of the stimulation window. The participants' responses could thus consist of 'yes, sure', 'yes, unsure', 'no, sure', and 'no, unsure'. The task was stopped after two sessions of 71 trials, or before in case of discomfort or other clinical constraints. As – upon waking from anesthesia – most participants did not use both confidence levels, confidence data was therefore not analyzed.

To keep the vibrotactile stimulus intensity around the detection threshold, we first conducted a rough threshold search by presenting a series of stimuli whose intensity decreased by steps of 5% until participants reported not feeling them anymore. Then, we presented a series of low-intensity stimuli whose intensities increased by steps of 5% until participants reported feeling them again. These procedures were repeated until the experimenter deemed the results satisfying. We took the average between the thresholds obtained during these procedures as a seed for the main task. During the main task, a 1-up/1-down adaptive staircase procedure (*Levitt, 1971*) ensured that the intensity was kept around the perceptual threshold by increasing the intensity by 5% after a miss trial and decreasing the intensity by 5% after a hit trial. Of note, the absolute stimulus intensity is not informative and cannot be compared across patients and sessions, as it varied according to different factors (e.g. the length of the cable or the manner with which the tactile stimulator was strapped onto the palm).

## Surgical procedure

STN or thalamus targets and trajectories were defined preoperatively using CranialSuite (Neurotargeting Inc, Nashville, TN, USA) based on MRI scans. Both targets were then defined with respect to the AC-PC (commissural) line using standard atlas-based methods and refined based on individual anatomy. The entry point was chosen approximately 2–3 cm from the midline and 1 cm anterior from the coronal suture and adjusted to individual anatomy in order to avoid traversing brain sulci, lateral ventricles, or the medial bridging veins. Scalp incisions and burr-hole drilling were performed under local (lidocaine) and general (propofol) anesthesia, and a microelectrode (FHC, ME, USA) was inserted through a guide cannula using a microdrive placed either on a Leksell frame (N=13 surgeries) or a 3D-printed mold (N=23 surgeries).

## Electrophysiology

Once the microelectrode reached the target brain structure (STN or thalamus), the speed of the microdrive was reduced, and neuronal activity was streamed to a loudspeaker, allowing the electrophysiologist to verify the depth of the preplanned trajectory. The main research task was initiated when a neuron showed stable activity for a few tens of seconds, and the anatomical localization was confirmed by the electrophysiologist. Recording depths were saved and used offline to define the anatomical localization (see Anatomical localization). Electrophysiological data were recorded from the 5 mm tip of the microelectrode, referenced to the guide cannula, and an adaptive line noise canceller was applied. Data were digitized either using a Guideline 4000 LP+ amplifier (FHC, ME,

USA) at 30 kHz (N=21 surgeries) or using a Guideline 5 amplifier (FHC, ME, USA) at 32 kHz and resampled offline to 30 kHz (N=14 surgeries).

## Anatomical localization

For 34/50 neurons, preoperative MRI and postoperative CT scans (co-registered in patient's native space using CranialSuite) were available to precisely reconstruct surgical trajectories and recording locations (for the remaining 16 neurons, localizations were based on neurosurgical planning and confirmed by electrophysiological recordings at various depths). Recording depths were inspected along the trajectories in patient's native space, projected onto an MNI-coordinate space and compared against the Ilinsky atlas (*Ilinsky et al., 2018*), which delineates distinct thalamic sub-territories based on a marker of γ-aminobutyric acid on sections of postmortem human brains.

## Behavioral analyses

We used R 4.1.2 (*R Development Core Team, 2013*) and the tidyverse (*Wickham et al., 2019*) package to analyze behavioral data. Permutation tests were performed by permuting hit and miss trials over 1000 iterations for each participant. Nonparametric p-values were estimated by counting the permutations for which the difference between hits and misses was higher in the observed compared to the shuffled data.

As titrating and keeping the vibrotactile stimulation intensity to the perceptual level after anesthesia was a challenging task, we took great care in keeping only the highest quality recordings. We estimated the trial-by-trial hit rate using a sliding window of 11 trials (for the first and last 5 trials, we mirrored trials to avoid border effects). Any trial with a hit rate out of the] 25, 75 [% range was removed from further analysis comparing hit to miss trials. If less than 10 hit and 10 miss trials were kept by this procedure, the session (and its corresponding neurons) was removed from subsequent analyses (13/48 sessions; 27%).

## Spike sorting and firing rate estimation

Each microelectrode recording was filtered between 300 and 3000 Hz and visually inspected. Artifacts such as cross-talk from the participants' vocal responses were marked and replaced by noise with a standard deviation matching the second pre- and post-artifact. We performed this procedure to avoid spuriously lowering the thresholds for neuronal spike detection. The timing of these artifactual epochs was saved in order to reject affected trials in later analyses. Neuronal spikes were detected and clustered using an online semiautomatic spike sorting algorithm (Osort) (*Rutishauser et al., 2006*). Each resulting cluster of neurons was inspected based on common metrics, such as spike waveform, percentage of inter-spike interval below 3 ms, signal-to-noise ratio, and power spectral densities, and possibly merged with other clusters. Finally, the resulting curated neurons were labeled as *putative single neuron* or *multiunit*, depending on the spike waveforms, peak amplitude distribution, and the percentage of inter-spike interval below 3 ms. Electrophysiological signals were realigned either to the onset of the 'go' cue (*Figure 2*) or to the onset of the stimulus (*Figures 3 and 4*), which was precisely obtained by applying a matched filter to a copy of the audio signal used to drive the vibrotactile stimulator we simultaneously recorded with the electrophysiological data. We estimated instantaneous firing rates using a sliding Gaussian kernel with a standard deviation of 40 ms and 1 ms steps. When displaying the resulting average firing rates over time, we estimated the standard error of the mean using a bootstrap procedure with 1000 resamplings.

## Statistical analyses

To deal with possibly nonparametric distributions, we used the Wilcoxon rank sum test or sign test instead of t-tests to test the differences between distributions. We used permutation tests instead of binomial tests to test whether a reported number of neurons could have been obtained by chance. For each analysis, we verified that the reported number of neurons could not have been obtained by chance by comparing this number to a null distribution using permutation tests (*Maris and Oostenveld, 2007*). For paired tests with respect to a baseline, we randomly flipped the sign of the difference between the firing rate during the trial and during the baseline (*sign permutation test* in the main text), and for unpaired tests, we randomly shuffled the conditions (i.e. a hit trial could be randomly assigned to a hit or a miss trial; *trial permutation test* in the main text). To obtain a p-value, we

compared the number of selective neurons to a null distribution obtained by randomly permuting the data 1000 times. This procedure allowed us to show that the number of selective neurons could not have been obtained by chance while controlling for multiple comparisons over time. Similarly, to test whether the proportion of neurons was different in the STN compared to the thalamus, we compared the absolute difference in the proportion of neurons in each anatomical location to a null distribution obtained by random permutations.

## Identification of selective neurons

To identify cue-selective neurons, we compared the number of spikes in a 500 ms baseline preceding the 'go' cue to the number of spikes in a 500 ms period following the offset of the 'go cue' using a two-tailed nonparametric sign test. Similarly, we identified task-responsive neurons by comparing the mean number of spikes in a 500 ms baseline preceding the 'go' cue to the mean number of spikes during the 2 s stimulation window and performing a permutation test. We compared the differences in the proportion of selective neurons in the STN and thalamus to the same differences observed in the shuffled data to assess its significance. Finally, we also compared the number of cue- and task-selective neurons to the same number observed in the shuffled data to assess whether the overlap was significant.

To identify sensory-selective neurons, we assumed that subcortical signatures of stimulus detection ought to be found early following its onset and looked for differences in the firing rates during the first 400 ms post-stimulus onset compared to a 300 ms pre-stimulus baseline. To correct for possible drifts occurring during the trial, we subtracted the average cue-locked activity from catch trials to the cue-locked activity of each stimulus-present trial before realigning to stimulus onset. We defined a cluster as a set of adjacent time points for which the firing rates were significantly different between hits and misses, as assessed by a nonparametric sign rank test. A putative neuron was considered sensory-selective when the length of a cluster was above 80 ms, corresponding to twice the standard deviation of the smoothing kernel used to compute the firing rate. Whether for the shuffled data or the observed data, if more than one cluster was obtained, we discarded all but the longest cluster. This permutation test allowed us to control for multiple comparisons across time and participants.

For perception-selective neurons, we looked for differences in the firing rates between hit and miss trials during the first 400 ms post-stimulus onset. We defined a cluster as a set of adjacent time points for which the firing rates were significantly different between hits and misses as assessed by a nonparametric Wilcoxon rank sum test. As for sensory-selective neurons, a putative neuron was considered perception-selective when the length of a cluster was above 80 ms, corresponding to twice the standard deviation of the smoothing kernel used to compute the firing rate, and we discarded all but the longest cluster.

To measure bimodal timings of effect latencies, we fitted a two-component Gaussian mixture distribution to the data in *Figure 5* by minimizing the mean square error with an interior-point method. We took the best of 20 runs with random initialization points and verified that the resulting mean square error was markedly (>4 times) better than using a single component.

## Acknowledgements

MP was supported by two Postdoc Mobility fellowships from the Swiss National Science Foundation (P2ELP3_187974; P400PM_199251). NF has received funding from the European Research Council (ERC) under the European Union's Horizon 2020 research and innovation program (grant agreement no. 803122). OB is supported by the Bertarelli Foundation, the Swiss National Science Foundation, and the European Science Foundation.

## Additional information

### Competing interests

Shuo Wang: Reviewing editor, *eLife*. Olaf Blanke: O.B. is cofounder and shareholder of Metaphysiks Engineering SA. O.B. is member of the board and shareholder of Mindmaze SA. The other authors declare that no competing interests exist.

## Funding

| Funder | Grant reference number | Author |
| --- | --- | --- |
| Swiss National Science Foundation | P2ELP3_187974 | Michael Pereira |
| Swiss National Science Foundation | P400PM_199251 | Michael Pereira |
| European Research Council | 803122 | Nathan Faivre |
| Bertarelli Foundation | | Olaf Blanke |
| Swiss National Science Foundation | | Olaf Blanke |
| European Science Foundation | | Olaf Blanke |

The funders had no role in study design, data collection and interpretation, or the decision to submit the work for publication.

### Author contributions

Michael Pereira, Conceptualization, Data curation, Formal analysis, Funding acquisition, Investigation, Writing – original draft, Writing – review and editing; Nathan Faivre, Conceptualization, Formal analysis, Funding acquisition, Investigation, Writing – original draft, Writing – review and editing; Fosco Bernasconi, Conceptualization, Data curation, Formal analysis, Investigation, Writing – original draft, Writing – review and editing; Nicholas Brandmeir, Jacob E Suffridge, Kaylee Tran, Peter Konrad, Investigation, Writing – review and editing; Shuo Wang, Resources, Writing – review and editing; Victor Finomore, Project administration, Writing – review and editing; Ali Rezai, Supervision, Funding acquisition, Project administration, Writing – review and editing; Olaf Blanke, Conceptualization, Supervision, Funding acquisition, Project administration, Writing – review and editing

### Author ORCIDs

Michael Pereira (ID) https://orcid.org/0000-0003-0778-674X
Nathan Faivre (ID) https://orcid.org/0000-0001-6011-4921
Shuo Wang (ID) https://orcid.org/0000-0003-2562-0225

### Ethics

The study was approved by the institutional review board of the West Virginia University Hospital (WVU02HSC17; #1709745061), and all participants provided written informed consent prior to any data collection.

Reviewer #1 (Public review): https://doi.org/10.7554/eLife.95272.3.sa1
Reviewer #2 (Public review): https://doi.org/10.7554/eLife.95272.3.sa2
Reviewer #3 (Public review): https://doi.org/10.7554/eLife.95272.3.sa3
Author response https://doi.org/10.7554/eLife.95272.3.sa4

# Additional files

### Supplementary files
MDAR checklist

### Data availability

The data and code necessary to replicate our results are available online (https://gitlab.com/michael.pereira/subcortical-ncc copy archived at *Pereira, 2025*).Further information and requests should be directed to and will be fulfilled by the lead contact, Michael Pereira (michael.pereira@univ-grenoble-alpes.fr).

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
