## [Editor Report · eLife Assessment]

This **important** study reports human single-neuron recordings in subcortical structures while participants performed a tactile detection task around the perceptual threshold. The study and the analyses are well conducted and provide **convincing** evidence that the thalamus and the subthalamic nucleus contain neurons whose activity correlates with the task, with stimulus presentation, and even with whether the stimulation is consciously detected or not. The study will be relevant for researchers interested in the role of subcortical structures in tactile perception and the neural correlates of consciousness.

---

## [Referee Report · Reviewer #1 (Public review)]

Summary:

A cortico-centric view is dominant in the study for the neural mechanisms of consciousness. This investigation represents the growing interest to understand how subcortical regions are involved in conscious perception. To achieve this, the authors engaged an ambitious and rare procedure in humans of directly recording from neurons in the subthalamic nucleus and thalamus. While participants were in surgery for the placement of deep brain stimulation devices for the treatment of essential tremor and Parkinson's disease, they were awakened and completed a perceptual-threshold tactile detection task. The authors identified individual neurons and analyzed single-unit activity corresponding with the task phases and tactile detection/perception. Among the neurons that were perception-responsive, the authors report changes in firing rate beginning ~150 milliseconds from the onset of the tactile stimulation. Curiously, the majority of the perception-responsive neurons had a higher firing rate for missed/not perceived trials. In summary, this investigation is a valuable addition to the growing literature on the role of subcortical regions in conscious perception.

Strengths:

The authors achieve the challenging task of recording human single-unit activity while participants performed a tactile perception task. The methods and statistics are clearly explained and rigorous, particularly for managing false positives and non-normal distributions. The results offer new detail at the level of individual neurons in the emerging recognition for the role of subcortical regions in conscious perception. Also, this study highlights the timing of neural activity linked to conscious perception (approximately 150 millisecond).

Weaknesses:

Due to constraints of testing with this patient population, a standard report-based detection task was administered. This type of task cannot fully exclude motor preparatory and post-perceptual processing as a factor that contributes to distinguishing between perceived versus not perceived stimuli. The authors show sensitivity to this issue by identifying task-selective neurons and their discussion of the results that refers to the confound of post-perceptual processing. Despite this limitation, the results are valuable for contributing to a growing body of literature on the subcortical neural mechanisms of consciousness.

---

## [Referee Report · Reviewer #2 (Public review)]

The authors have examined subpopulations of individual neurons recorded in the thalamus and subthalamic nucleus (STN) of awake humans performing a simple cognitive task. They have carefully designed their task structure to minimize motor components that could confound their analyses in these subcortical structures, particularly given that the data was collected from patients with Parkinson's disease (PD) and essential tremor (ET). The recorded data represents a valuable contribution to the field. Pereira et al. conclude that their single-neuron recordings indicate task-related activity that is purportedly distinct from previously identified sensory signals.

Despite the significance of the dataset, important limitations arise due to the small number of recorded neurons relative to the high number of participants. That raises concerns about the generalizability of the conclusions drawn from the study.

(1) While I support the work conducted by the authors and their efforts to improve the manuscript, the number of significant neurons is considerably lower than the number of participants studied-approximately 8 neurons, compared to 32 participants. This low number of neurons involved in encoding raises concerns about the strength of the conclusions drawn.

(2) Additionally, the authors state that participants do not need to perform a motor execution, yet they are required to communicate their response verbally. This presents a contradiction, as speech involves the activation of facial muscles, and previous studies have shown that neuronal activity in the ventral premotor cortex can encode such movements in humans (Willet et al., Nature 2023). Clarifying this point would strengthen the argument and ensure consistency in the interpretation of results.

(3) One way to improve the study is to analyze the local field potentials (LFPs) recorded alongside the spikes. By examining different LFP components, particularly the beta band (Haegens et al., PNAS 2011), it may be possible to identify consistent modulation across the 32 recorded participants. This approach could provide additional support for the study's conclusions and help clarify the role of neural activity in the observed phenomena.

---

## [Referee Report · Reviewer #3 (Public review)]

Summary:

This important study relies on a rare dataset: intracranial recordings within the thalamus and the subthalamic nucleus in awake humans, while they were performing a tactile detection task. This procedure allowed the authors to identify a small but significant proportion of individual neurons, in both structures, whose activity correlated with the task (e.g. their firing rate changed following the audio cue signalling the start of a trial) and/or with the stimulus presentation (change in firing rate around 200 ms following tactile stimulation) and/or with participant's reported subjective perception of the stimulus (difference between hits and misses around 200 ms following tactile stimulation). Whereas most studies interested in the neural underpinnings of conscious perception focus on cortical areas, these results suggest that subcortical structures might also play a role in conscious perception, notably tactile detection.

Strengths:

There are two strongly valuable aspects in this study that make the evidence convincing and even compelling. First, these type of data are exceptional, the authors could have access to subcortical recordings in awake and behaving humans during surgery. Additionally, the methods are solid. The behavioral study meets the best standards of the domain, with a careful calibration of the stimulation levels (staircase) to maintain them around detection threshold, and additional selection of time intervals where the behavior was stable. The authors also checked that stimulus intensity was the same on average for hits and misses within these selected periods, which warrants that the effects of detection that are observed here are not confounded by stimulus intensity. The neural data analysis is also very sound and well conducted. The statistical approach complies to current best practices, although I found that, on some instances, it was not entirely clear which type of permutations had been performed, and I would advocate for more clarity in these instances. Globally, the figures are nice, clear and well presented. I appreciated the fact that the precise anatomical location of the neurons was directly shown in each figure.

Weaknesses:

The results rely on a small number of neurons; it is only the beginning of this exploration! Figure S5 is important for observing the variety of ways the neurons' activity correlated with either stimulus presence, or perception, or both. Interpretations are still very open on these different profiles.

---

## [Author Response]

The following is the authors’ response to the original reviews

**Public Reviews:**

**Reviewer #1 (Public Review):**
Summary:A cortico-centric view is dominant in the study of the neural mechanisms of consciousness. This investigation represents the growing interest in understanding how subcortical regions are involved in conscious perception. To achieve this, the authors engaged in an ambitious and rare procedure in humans of directly recording from neurons in the subthalamic nucleus and thalamus. While participants were in surgery for the placement of deep brain stimulation devices for the treatment of essential tremor and Parkinson's disease, they were awakened and completed a perceptual-threshold tactile detection task. The authors identified individual neurons and analyzed single-unit activity corresponding with the task phases and tactile detection/perception. Among the neurons that were perception-responsive, the authors report changes in firing rate beginning ~150 milliseconds from the onset of the tactile stimulation. Curiously, the majority of the perception-responsive neurons had a higher firing rate for missed/not perceived trials. In summary, this investigation is a valuable addition to the growing literature on the role of subcortical regions in conscious perception.Strengths:The authors achieved the challenging task of recording human single-unit activity while participants performed a tactile perception task. The methods and statistics are clearly explained and rigorous, particularly for managing false positives and non-normal distributions. The results offer new detail at the level of individual neurons in the emerging recognition of the role of subcortical regions in conscious perception.

We thank the reviewer for their positive comments.

Weaknesses:"Nonetheless, it remains unknown how the firing rate of subcortical neurons changes when a stimulus is consciously perceived." (lines 76-77) The authors could be more specific about what exactly single-unit recordings offer for interrogating the role of subcortical regions in conscious perception that is unique from alternative neural activity recordings (e.g., local field potential) or recordings that are used as proxies of neural activity (e.g., fMRI).

We agree with the reviewer that the contribution of micro-electrode recordings was not sufficiently put forward in our manuscript. We added the following sentences to the discussion, when discussing the multiple types of neurons we found:

Single-unit recordings provide a much higher temporal resolution than functional imaging, which helps assess how the neural correlates of consciousness unfold over time. Contrary to local field potentials, single-unit recordings can expose the variety of functional roles of neurons within subcortical regions, thereby offering a potential for a better mechanistic understanding of perceptual consciousness.

Related comment for the following excerpts:"After a random delay ranging from 0.5 to 1 s, a "respond" cue was played, prompting participants to verbally report whether they felt a vibration or not. Therefore, none of the reported analyses are confounded by motor responses." (lines 97-99)."These results show that subthalamic and thalamic neurons are modulated by stimulus onset, irrespective of whether it was reported or not, even though no immediate motor response was required." (lines 188190)."By imposing a delay between the end of the tactile stimulation window and the subjective report, we ensured that neuronal responses reflected stimulus detection and not mere motor responses." (lines 245247).It is a valuable feature of the paradigm that the reporting period was initiated hundreds of milliseconds after the stimulus presentation so that the neural responses should not represent "mere motor responses". However, verbal report of having perceived or not perceived a stimulus is a motor response and because the participants anticipate having to make these reports before the onset of the response period, there may be motor preparatory activity from the time of the perceived stimulus that is absent for the not perceived stimulus. The authors show sensitivity to this issue by identifying task-selective neurons and their discussion of the results that refer to the confound of post-perceptual processing. Still, direct treatment of this possible confound would help the rigor of the interpretation of the results.

We agree with the reviewer that direct treatment would have provided the best control. One way to avoid motor preparation is to only provide the stimulus-effector mapping after the stimulus presentation (Bennur & Gold, 2011; Twomey et al., 2016; Fang et al., 2024). Other controls to avoid post-perceptual processing used in consciousness research consist of using no-report paradigms (Tsuchiya et al., 2015) as we did in previous studies (Pereira et al., 2021; Stockart et al., 2024). Unfortunately, neither of these procedures was feasible during the 10 minutes allotted for the research task in an intraoperative setting with auditory cues and vocal responses. We would like to highlight nonetheless that the effects we report are shortlived and incompatible with sustained motor preparation activity.

We added the following sentence to the discussion:

Future studies ruling out the presence of motor preparation triggered by perceived stimuli (Bennur & Gold, 2011; Fang et al., 2024; Twomey et al., 2016) and verifying that similar neuronal activity occurs in the absence of task-demands (no-reports; Tsuchiya et al., 2015) or attention (Wyart & Tallon-Baudry, 2008) will be useful to support that subcortical neurons contribute specifically to perceptual consciousness.

"When analyzing tactile perception, we ensured that our results were not contaminated with spurious behavior (e.g. fluctuation of attention and arousal due to the surgical procedure)." (lines 118-117).Confidence in the results would be improved if the authors clarified exactly what behaviors were considered as contaminating the results (e.g., eye closure, saccades, and bodily movements) and how they were determined.

This sentence was indeed unclear. It introduced the trial selection procedure we used to compensate for drifts in the perceptual threshold, which can result from fluctuations in attention or arousal. We modified the sentence, which now reads:

When analyzing tactile perception, we ensured that our results were not contaminated by fluctuating attention and arousal due to the surgical procedure. Based on objective criteria, we excluded specific series of trials from analyses and focused on time windows for which hits and misses occurred in commensurate proportions (see Methods).

During the recordings, the experimenter stood next to the patients and monitored their bodily movements, ensuring they did not close their eyes or produce any other bodily movements synchronous with stimulus presentation.

The authors' discussion of the thalamic neurons could be more precise. The authors show that only certain areas of the thalamus were recorded (in or near the ventral lateral nucleus, according to Figure S3C). The ventral lateral nucleus has a unique relationship to tactile and motor systems, so do the authors hypothesize these same perception-selective neurons would be active in the same way for visual, auditory, olfactory, and taste perception? Moreover, the authors minimally interpret the location of the task, sensory, and perception-responsive neurons. Figure S3 suggests these neurons are overlapping. Did the authors expect this overlap and what does it mean for the functional organization of the ventral lateral nucleus and subthalamic nucleus in conscious perception?

These are excellent questions, the answers to which we can only speculate. In rodents, the LT is known as a hub for multisensory processing, as over 90% of LT neurons respond to at least two sensory modalities (for a review, see Yang et al., 2024). Yet, no study has compared how LT neurons in rodents encode perceived and nonperceived stimuli across modalities. Evidence in humans is scarce, with only a few studies documenting supramodal neural correlates of consciousness at the cortical level with noninvsasive methods (Noel et al., 2018; Sanchez et al., 2020; Filimonov et al., 2022). We now refer to these studies in the revised discussion: Moreover, given the prominent role of the thalamus in multisensory processing, it will be interesting to assess if it is specifically involved in tactile consciousness or if it has a supramodal contribution, akin to what is found in the cortex (Noel et al., 2018; Sanchez et al., 2020; Filimonov et al., 2022).

Concerning the anatomical overlap of neurons, we could not reconstruct the exact locations of the DBS tracts for all participants. Because of the limited number of recorded neurons, we preferred to refrain from drawing strong conclusions about the functional organization of the ventral lateral nucleus.

"We note that, 6 out of 8 neurons had higher firing rates for missed trials than hit trials, although this proportion was not significant (binomial test: p = 0.145)." (lines 215-216).It appears that in the three example neurons shown in Figure 4, 2 out of 3 (#001 and #068) show a change in firing rate predominantly for the missed stimulations. Meanwhile, #034 shows a clear hit response (although there is an early missed response - decreased firing rate - around 150 ms that is not statistically significant). This is a counterintuitive finding when compared to previous results from the thalamus (e.g., local field potentials and fMRI) that show the opposite response profile (i.e., missed/not perceived trials display no change or reduced response relative to hit/perceived trials). The discussion of the results should address this, including if these seemingly competing findings can be rectified.

We thank the reviewer for pointing out this limitation of the discussion. We avoided putting too much emphasis on these aspects due to the limited number of perception-selective neurons. Although subcortical connectivity models would predict that neurons in the thalamus should increase their firing rate for perceived stimuli, we were not surprised to see this heterogeneity as we had previously found neurons decreasing their firing rates for missed stimuli in the posterior parietal cortex (Pereira et al., 2021). We answer these points in response to the reviewer’s last comment below on the latencies of the effects.

The authors report 8 perception-responsive neurons, but there are only 5 recording sites highlighted (i.e., filled-in squares and circles) in Figures S3C and 4D. Was this an omission or were three neurons removed from the perception-responsive analysis?

Unfortunately, we could not obtain anatomical images for all participants. This information was present in the methods section, although not clearly enough:

For 34 / 50 neurons, preoperative MRI and postoperative CT scans (co-registered in patient native space using CranialSuite) were available to precisely reconstruct surgical trajectories and recording locations (for the remaining 16 neurons, localizations were based on neurosurgical planning and confirmed by electrophysiological recordings at various depths).

Therefore, we added the following sentence in Figures 2, 3, 4 and S3.

[...] for patients for which we could obtain anatomical images.

Could the authors speak to the timing of the responses reported in Figure 4? The statistically significant intervals suggested both early (~160-200ms) to late responses (~300ms). Some have hypothesized that subcortical regions are early - ahead of cortical activation that may be linked with conscious perception. Do these results say anything about this temporal model for when subcortical regions are active in conscious perception?

We agree that response timing could have been better described. We performed a new analysis of the latencies at which our main effects were observed. This analysis revealed the existence of the two clusters mentioned by the reviewer very clearly. We now include this analysis in a new Figure 5 in the revised manuscript.

We also performed a new analysis to support the existence of bimodal distributions and quantified the latencies. We added this text to the result section:

We note that the timings of sensory and perception effects in Figures 3 and 4 showed a bimodal distribution with an early cluster (149 ms for sensory neurons; 121 ms for perception neurons; c.f. methods) and a later cluster (330 ms for sensory neurons; 315 ms for perception neurons; Figure 5). and this section to the methods:

To measure bimodal timings of effect latencies, we fitted a two-component Gaussian mixture distribution to the data in Figure 5 by minimizing the mean square error with an interior-point method. We took the best of 20 runs with random initialization points and verified that the resulting mean square error was markedly (> 4 times) better than using a single component.

We updated the discussion, including the points made in the comment about higher activity for missed stimuli (above):

The early cluster’s average timing around 150 ms post-stimulus corresponds to the onset of a putative cortical correlate of tactile consciousness, the somatosensory awareness negativity (Dembski et al., 2021). Similar electroencephalographic markers are found in the visual and auditory modality. It is unclear, however, whether these markers are related to perceptual consciousness or selective attention (Dembski et al., 2021). The later cluster is centered around 300 ms and could correspond to a well known electroencephalographic marker, the P3b (Polich, 2007) whose association with perceptual consciousness has been questioned (Pitts et al., 2014; Dembski et al., 2021) although brain activity related to consciousness has been observed at similar timing even in the absence of report demands (Sergent et al., 2021; Stockart et al., 2024). It is also important to note that these clusters contain neurons with both increased and decreased firing rates following stimulus onset, similar to what was observed previously in the posterior parietal cortex (Pereira et al., 2021).

**Reviewer #2 (Public Review):**
The authors have studied subpopulations of individual neurons recorded in the thalamus and subthalamic nucleus (STN) of awake humans performing a simple cognitive task. They have carefully designed their task structure to eliminate motor components that could confound their analyses in these subcortical structures, given that the data was recorded in patients with Parkinson's Disease (PD) and diagnosed with an Essential Tremor (ET). The recorded data represents a promising addition to the field. The analyses that the authors have applied can serve as a strong starting point for exploring the kinds of complex signals that can emerge within a single neuron's activity. Pereira et. al conclude that their results from single neurons indicate that task-related activity occurs, purportedly separate from previously identified sensory signals. These conclusions are a promising and novel perspective for how the field thinks about the emergence of decisions and sensory perception across the entire brain as a unit.

We thank the reviewer for these positive comments.

Despite the strength of the data that was obtained and the relevant nature of the conclusions that were drawn, there are certain limitations that must be taken into consideration:(1) The authors make several claims that their findings are direct representations of consciousnessidentifiable in subcortical structures. The current context for consciousness does not sufficiently define how the consciousness is related to the perceptual task.

This is indeed a complex issue in all studies concerned with perceptual consciousness and we were careful not to make such “direct” claims. Instead, we used the state-of-the-art tools available to study consciousness (see below) and only interpreted our findings with respect to consciousness in the discussion. For example, in the abstract, our claim is that “Our results provide direct neurophysiological evidence of the involvement of the subthalamic nucleus and the thalamus for the detection of vibrotactile stimuli, thereby calling for a less cortico-centric view of the neural correlates of consciousness.”

In brief, first, we used near-threshold stimuli which allowed us to contrast reported vs. unreported trials while keeping the physical properties of the stimulus comparable. Second, we used subjective reports without incentive for participants to be more conservative or liberal in their response (e.g. through reward). Third, we introduced a random delay before the responses to limit confounding effects due to the report. We also acknowledged that “... it will be important in future studies to examine if similar subcortical responses are obtained when stimuli are unattended (Wyart & Tallon-Baudry, 2008), task-irrelevant (Shafto & Pitts, 2015), or when participants passively experience stimuli without the instruction to report them (i.e., no-report paradigms) (Tsuchyia et al., 2015)”. This last sentence now reads (to address a point made by Reviewer 1 about motor preparation):

Future studies ruling out the presence of motor preparation triggered by perceived stimuli (Bennur & Gold, 2011; Fang et al., 2024; Twomey et al., 2016) and verifying that similar neuronal activity occurs in the absence of task-demands (no-reports; Tsuchiya et al., 2015) or attention (Wyart & Tallon-Baudry, 2008) will be useful to support that subcortical neurons contribute specifically to perceptual consciousness.

(2) The current work would benefit greatly from a description and clarification of what all the neurons thathave been recorded are doing. The authors' criteria for selecting subpopulations with task-relevant activity are appropriate, but understanding the heterogeneity in a population of single neurons is important for broader considerations that are being studied within the field.

We followed the reviewer’s suggestions and added new results regarding the latencies of the reported effects (new Figure 5). We also now show firing rates for hits, misses and overall sensory activity (hits and misses combined) for all perception-selective or sensory-selective (when behavior was good enough; Figure S5). Although a more detailed characterization of the heterogeneity of the neurons identified would have been relevant, it seems beyond the scope of the present study, especially given the relatively small number of neurons we identified, as well as the relative simplicity of the paradigm imposed by the clinical context in which we worked.

(3) The authors have omitted a proper set of controls for comparison against the active trials, forexample, where a response was not necessary. Please explain why this choice was made and what implications are necessary to consider.

We had mentioned this limitation in the discussion: Nevertheless, it will be important in future studies to examine if similar subcortical responses are obtained when stimuli are unattended (Wyart & TallonBaudry, 2008), task-irrelevant (Shafto & Pitts, 2015), or when participants passively experience stimuli without the instruction to report them (i.e., no-report paradigms) (Tsuchyia et al., 2015). We agree that such a control would have been relevant, but this was not feasible during the 10 minutes allotted for the research task in an intraoperative setting. These constraints are both clinical, to minimize discomfort for patients and practical, as is difficult to track neurons in an intraoperative setting for more than 10 minutes.

We added a sentence to this effect in the discussion.

**Reviewer #3 (Public Review):**
Summary:This important study relies on a rare dataset: intracranial recordings within the thalamus and the subthalamic nucleus in awake humans, while they were performing a tactile detection task. This procedure allowed the authors to identify a small but significant proportion of individual neurons, in both structures, whose activity correlated with the task (e.g. their firing rate changed following the audio cue signalling the start of a trial) and/or with the stimulus presentation (change in firing rate around 200 ms following tactile stimulation) and/or with participant's reported subjective perception of the stimulus (difference between hits and misses around 200 ms following tactile stimulation). Whereas most studies interested in the neural underpinnings of conscious perception focus on cortical areas, these results suggest that subcortical structures might also play a role in conscious perception, notably tactile detection.Strengths:There are two strongly valuable aspects in this study that make the evidence convincing and even compelling. First, these types of data are exceptional, the authors could have access to subcortical recordings in awake and behaving humans during surgery. Additionally, the methods are solid. The behavioral study meets the best standards of the domain, with a careful calibration of the stimulation levels (staircase) to maintain them around the detection threshold, and an additional selection of time intervals where the behavior was stable. The authors also checked that stimulus intensity was the same on average for hits and misses within these selected periods, which warrants that the effects of detection that are observed here are not confounded by stimulus intensity. The neural data analysis is also very sound and well-conducted. The statistical approach complies with current best practices, although I found that, in some instances, it was not entirely clear which type of permutations had been performed, and I would advocate for more clarity in these instances. Globally the figures are nice, clear, and well presented. I appreciated the fact that the precise anatomical location of the neurons was directly shown in each figure.

We thank the reviewer for this positive evaluation.

Weaknesses:Some clarification is needed for interpreting Figure 3, top rows: in my understanding the black curve is already the result of a subtraction between stimulus present trials and catch trials, to remove potential drifts; if so, it does not make sense to compare it with the firing rate recorded for catch trials.

The black curve represents the firing rate without any subtraction. We only subtracted the firing rates of catch trials in the statistical procedure, as the reviewer noted, to remove potential drift. We added (before baseline correction) to the legend of Figure 3.

I also think that the article could benefit from a more thorough presentation of the data and that this could help refine the interpretation which seems to be a bit incomplete in the current version. There are 8 stimulus-responsive neurons and 8 perception-selective neurons, with only one showing both effects, resulting in a total of 15 individual neurons being in either category or 13 neurons if we exclude those in which the behavior is not good enough for the hit versus miss analysis (Figure S4A). In my opinion, it should be feasible to show the data for all of them (either in a main figure, or at least in supplementary), but in the present version, we get to see the data for only 3 neurons for each analysis. This very small selection includes the only neuron that shows both effects (neuron #001; which is also cue selective), but this is not highlighted in the text. It would be interesting to see both the stimulus-response data and the hit versus miss data for all 13 neurons as it could help develop the interpretation of exactly how these neurons might be involved in stimulus processing and conscious perception. This should give rise to distinct interpretations for the three possible categories. Neurons that are stimulus-responsive but not perception-selective should show the same response for both hits and misses and hence carry out indifferently conscious and unconscious responses. The fact that some neurons show the opposite pattern is particularly intriguing and might give rise to a very specific interpretation: if the neuron really doesn't tend to respond to the stimulus when hits and misses are put together, it might be a neuron that does not directly respond to the stimulus, but whose spontaneous fluctuations across trials affect how the stimulus is perceived when they occur in a specific time window after the stimulus. Finally, neuron #001 responds with what looks like a real burst of evoked activity to stimulation and also shows a difference between hits and misses, but intriguingly, the response is strongest for misses. In the discussion, the interesting interpretation in terms of a specific gating of information by subcortical structures seems to apply well to this last example, but not necessarily to the other categories.

We now provide a supplementary Figure showing firing rates for hits, misses and the combination of both. The reviewer’s analysis about whether a perception-selective neuron also has to respond to the stimulus to be involved in gating is interesting. With more data, a finer characterization of these neurons would have been possible. In our study, it is possible that more neurons have similar characteristics as #001 (e.g. #032, #062, #068) but do not show a significant difference with respect to baseline when both hits and misses are considered. We now avoid interpreting null effects, especially considering the low number of trials with near-threshold detection behavior we could collect in 10 minutes.

We also realized that we had not updated Figure S7 after the last revision in which we had corrected for possible drifts to obtain sensory-selective neurons. The corrected panel A is provided below.

**Recommendations for the authors:**

**Reviewer #1 (Recommendations For The Authors):**
It appears that the correct rejection was low for most participants. It would improve interpretation of the behavioral results if correct rejection was shown as a rate (i.e., # of correct rejection trials / total number of no stimulus/blank trials) rather than or in addition to reporting the number of correct rejection trials (Figure 1C).

We added the following figure to the supplementary information.

The axis tick marks in Figure 5A late versus early are incorrect (appears the axis was duplicated).

Thank you for spotting this, it has been corrected.

**Reviewer #2 (Recommendations For The Authors):**
We would like to congratulate the authors on this strongly supported contribution to the field. The manuscript is well-written, although a little bit too concise in sections. See the following comments for the methods that could benefit the present conclusions:

Thank you for these suggestions that we believe improved our interpretations.

Major Points(1) The subpopulations of neurons that are considered are small, but it is not a confounding issue for the conclusions drawn. However, the behavior of the neurons that were excluded should be considered by calculating the percentage of neurons that are selective for the distinct parameters, as a function of time. This would greatly strengthen the understanding of what can be observed in the two subcortical structures.

We thank the reviewer for this suggestion. We performed a new analysis of the latencies at which our main effects were observed. This analysis revealed the existence of two clusters, as shown in the new Figure 5 copied below

We also performed a new analysis to support the existence of bimodal distributions and quantified the latencies. We added this text to the result section:

We note that the timings of sensory and perception effects in Figures 3 and 4 showed a bimodal distribution with an early cluster (149 ms for sensory neurons; 121 ms for perception neurons; c.f. methods) and a later cluster (330 ms for sensory neurons; 315 ms for perception neurons; Figure 5). and this section to the methods:

To measure bimodal timings of effect latencies, we fitted a two-component Gaussian mixture distribution to the data in Figure 5 by minimizing the mean square error with an interior-point method. We took the best of 20 runs with random initialization points and verified that the resulting mean square error was markedly (> 4 times) better than using a single component.

We also updated the discussion:

The early cluster’s average timing around 150 ms post-stimulus corresponds to the onset of a putative cortical correlate of tactile consciousness, the somatosensory awareness negativity (Dembski et al., 2021). Similar electroencephalographic markers are found in the visual and auditory modality. It is unclear, however, whether these markers are related to perceptual consciousness or selective attention (Dembski et al., 2021). The later cluster is centered around 300 ms and could correspond to a well known electroencephalographic marker, the P3b (Polich, 2007) whose association with perceptual consciousness has been questioned (Pitts et al., 2014; Dembski et al., 2021) although brain activity related to consciousness has been observed at similar timing even in the absence of report demands (Sergent et al., 2021; Stockart et al., 2024). It is also important to note that these clusters contain neurons with both increased and decreased firing rates following stimulus onset, similar to what was observed previously in the posterior parietal cortex (Pereira et al., 2021).

(2) We highly recommend that the authors consider employing some analysis that decodes therepresentations observable in the activity of individual neurons as a function of time (e.g. Shannon's Mutual Information). This would reinforce and emphasize the most relevant conclusions.

We thank the reviewers for this suggestion. Unfortunately, such methods would require many more trials than what we were able to collect in the 10-minute slots available in the operating room.

(3) Although there are small populations recorded in each of the two subcortical structures, they aresufficient to attempt a study using population dynamics (primarily, PCA can still work with smaller populations). Given the broad range of dynamics that are observed in a population of single units typically involved in decision-making, it would be interesting to consider whether heterogeneity is a hallmark of decision-making, and trying to summarize the variance in the activity of the entire population should provide a certain understanding of the cue-selective versus the perception-selective qualities, as an example.

We now present all 13 neurons that were sensory- or perception-selective for which we had good enough behavior to show hit vs. miss differences in Supplementary Figure S5. Although population-level analyses would be relevant, they are not compatible with the number of neurons we identified.

(4) A stronger presentation of what the expectations are for the results would also benefit theinterpretability of the manuscript when added to the introduction and discussion sections.

Due to the scarcity of single-neuron data related to perceptual consciousness, especially in the subcortical structures we explored, our prior expectations did not exceed finding perception-selective neurons. We would prefer to avoid refining these expectations post-hoc.

Minor Comments(1) Add the shared overlap between differently selective neurons explicitly in the manuscript.

We added this information at the end of the results section.

(2) Add a consideration in the methods of why the Wilcoxon test or permutation test was selected forseparate uses. How do the results compare?

Sorry for this misunderstanding. We clarified this in revised methods:

To deal with possibly non-parametric distributions, we used Wilcoxon rank sum test or sign test instead of t-tests to test differences between distributions. We used permutation tests instead of Binomial tests to test whether a reported number of neurons could have been obtained by chance.

**Reviewer #3 (Recommendations For The Authors):**
Suggestions for improved or additional experiments, data or analysis:As suggested already in the public review, it might be worth showing all 13 neurons with either stimulusresponsive or perception-selective behaviour and, based on that, deepen the potential interpretation of the results for the different categories.

We agree that this information improves the understanding of the underlying data and this addition was also proposed by reviewer 2. We added it in a new supplementary Figure S5.

Recommendations for improving the writing and presentationAs mentioned in the public review, I think Figure 3 needs clarification. I found that, in some instances, it was not entirely clear which type of analyses or permutation tests had been performed, and I would advocate for more clarity in these instances. For example:Page 6 line 146 "permuting trial labels 1000 times": do you mean randomly attributing a trial to aneuron? Or something else?

We agree that this was somewhat unclear. We modified the sentence to:

permuting the sign of the trial-wise differences

We now define a sign permutation test for paired tests and a trial permutation test for two-sample tests in the methods and specify which test was used in the maintext.

Page 7, neurons which have their firing rate modulated by the stimulus: I think you ought to be moreexplicit about the analysis so that we grasp it on the first read. To understand what is shown in Figure 3 I had to go back and forth between the main text and the method, and I am still not sure I completely understood. You compare the firing rate in sliding windows following stimulus onset with the mean firing rate during the 300ms baseline. Sliding windows are between 0 and 400 ms post-stim (according to methods ?) and a neuron is deemed responsive if you find at least one temporal cluster that shows a significant difference with baseline activity (using cluster permutation). Is that correct? Either way, I would recommend being a bit more precise about the analysis that was carried out in the main text, so that we only need to refer to methods when we need specialized information.

We agree that the methods section was unclear. We re-wrote the following two paragraphs:

To identify sensory-selective neurons, we assumed that subcortical signatures of stimulus detection ought to be found early following its onset and looked for differences in the firing rates during the first 400 ms post-stimulus onset compared to a 300 ms pre-stimulus baseline. To correct for possible drifts occurring during the trial, we subtracted the average cue-locked activity from catch trials to the cuelocked activity of each stimulus-present trials before realigning to stimulus onset. We defined a cluster as a set of adjacent time points for which the firing rates were significantly different between hits and misses, as assessed by a non-parametric sign rank test. A putative neuron was considered sensory-selective when the length of a cluster was above 80 ms, corresponding to twice the standard deviation of the smoothing kernel used to compute the firing rate. Whether for the shuffled data or the observed data, if more than one cluster was obtained, we discarded all but the longest cluster. This permutation test allowed us to control for multiple comparisons across time and participants.

For perception-selective neurons, we looked for differences in the firing rates between hit and miss trials during the first 400 ms post-stimulus onset. We defined a cluster as a set of adjacent time points for which the firing rates were significantly different between hits and misses as assessed by a nonparametric Wilcoxon rank sum test. As for sensory-selective neurons, a putative neuron was considered perception-selective when the length of a cluster was above 80 ms, corresponding to twice the standard deviation of the smoothing kernel used to compute the firing rate and we discarded all but the longest cluster.

Minor points:Figure 3: inset showing action potentials, please also provide the time scale (in the legend for example), so that it's clear that it is not commensurate with the firing rate curve below, but rather corresponds to the dots of the raster plot.

We added the text ”[...], duration: 2.5 ms” in Figures 2, 3, and 4.

Line 210: I recommend: “we found 8 neurons [...] showing a significant difference *between hits and misses* after stimulus onset."

We made the change.

Top of page 9, the following sentence is misleading “This result suggests that neurons in these two subcortical structures have mostly different functional roles ; this could read as meaning that functional roles are different between the two structures. Probably what you mean is rather something along this line : “these two subcortical structures both contain neurons displaying several different functional roles”

Changed.

Line 329: remove double “when”

We made the change, thank you for spotting this.